# Rapid quantification of miRNAs using dynamic FRET-FISH

Juyoung Kim [1], Chanshin Kang [1], Soochul Shin[1✉] & Sungchul Hohng [1✉]

MicroRNAs (miRNAs) are short regulatory RNAs that control gene expression at the post-transcriptional level. Various miRNAs playing important roles in cancer development are emerging as promising diagnostic biomarkers for early cancer detection. Accurate miRNA detection, however, remains challenging because they are small and highly homologous. Recently developed miRNA detection techniques based on single-molecule imaging enabled highly specific miRNA quantification without amplification, but the time required for these techniques to detect a single miRNA was larger than 10 minutes, making rapid profiling of numerous miRNAs impractical. Here we report a rapid miRNA detection technique, dynamic FRET-FISH, in which single-molecule imaging at high probe concentrations and thus high-speed miRNA detection is possible. Dynamic FRET-FISH can detect miRNAs in 10 s at 1.2 µM probe concentration while maintaining the high-specificity of single-nucleotide discrimination. We expect dynamic FRET-FISH will be utilized for early detection of cancers by profiling hundreds of cancer biomarkers in an hour.

[1] Department of Physics and Astronomy, and Institute of Applied Physics, Seoul National University, Seoul, Republic of Korea. ✉email: tlstncjf@snu.ac.kr; shohng@snu.ac.kr

MicroRNAs (miRNAs) are small (~22 nt) non-coding RNAs that regulate gene expression via post-transcriptional gene silencing[1]. Accurate profiling of miRNA expression is important in medicine as well as biology. Dysregulation of miRNAs is related to cell malfunction and disease development[2]. A large number of miRNAs are found to be dysregulated in a broad spectrum of cancers in a disease-specific fashion[3]. Stable and diagnostically useful miRNAs such as miR-20a[4,5] for lung cancer and miR-421[6,7] for gastric cancer have been detected in blood, making them ideal biomarkers for early cancer detection[8,9].

Due to their small size and the existence of highly homologous family members, miRNA detection presents tough challenges to conventional RNA detection technique such as microarray, polymerase chain reaction (PCR), and next-generation sequencing (NGS); their performances fall short of the diagnostic expectations in terms of either specificity, sensitivity, or quantification accuracy. The challenges were recently addressed by amplification-free miRNA detection techniques based on single-molecule fluorescence imaging called SiMREPS (Single Molecule Recognition through Equilibrium Poisson Sampling)[10] and Ago-FISH (Argonaute-based Fluorescence In Situ Hybridization)[11], providing reliable ways to accurately profile various miRNAs with highly-specific single-nucleotide discrimination. SiMREPS, a miRNA detection method inspired by the super-resolution imaging technique, DNA-PAINT (DNA-based point accumulation for imaging in nanoscale topography)[12], used the transient binding of fluorescently-labeled short DNA probes to surface-immobilized target miRNAs for miRNA detection, and demonstrated a highly-confident miRNA detection based on binding kinetics analysis with high specificity of 500-fold discrimination between single-nucleotide polymorphisms. Ago-FISH, another miRNA detection technique inspired by DNA-PAINT, uses the transient binding of Argonaute-loaded DNA probes instead of bare DNA probes to obtain 20-fold increased probe binding rate compared to SiMREPS at the same probe concentrations while maintaining the similar high target specificity of SiMREPS. The quantification accuracies of SiMREPS and Ago-FISH were also superior to the conventional techniques such as NGS and PCR because no nonlinear amplification step was required for miRNA quantification. Recently other amplification-free miRNA detection techniques based on target-probe hybridization have been developed[13–15], but they could not demonstrate such a high target specificity of SiMREPS and Ago-FISH because the techniques required a stable binding of probes to target miRNAs like the conventional microarray technique. A potential disadvantage of SiMREPS and Ago-FISH, however, is that they should detect different miRNAs sequentially, making the time required for multiple miRNA profiling linearly increase with the number of profiled miRNAs. Although SiMREPS and Ago-FISH can detect a single miRNA much faster (10 min) than PCR (several hours), this advantage disappears when hundreds of miRNAs should be profiled at once.

Here, we introduce a high-speed single-molecule miRNA detection technique, dynamic FRET-FISH (FRET-based Fluorescence In Situ Hybridization). Compared to SiMREPS and Ago-FISH, it has 60-fold increased detection speed while maintaining both the high target specificity and the high quantification accuracy of SiMREPS and Ago-FISH.

## Results

The single-molecule miRNA detection techniques, SiMREPS and Ago-FISH, were based on monitoring of repetitive binding of fluorophore-labeled DNA probes. It was conceivable to increase the probe binding rate, and thus decrease the miRNA detection time by increasing the probe concentration. This simple scheme, however, could not work because the background noise coming from floating fluorescent probes increased with the probe concentration, overwhelming single-molecule fluorescence signals coming from target-binding probes at probe concentrations higher than tens of nanomolar[16,17]. A key idea of dynamic FRET-FISH is to reduce the background fluorescence by using FRET (Fluorescence Resonance Energy Transfer). To be specific, we use Alexa-488 and Cy3-labeled DNAs as donor probes and Cy5-labeled DNA as an acceptor probe so that a high FRET state occurs only when both donor and acceptor probes simultaneously bind to the same target miRNA (Fig. 1a). Since only acceptor signals due to FRET are detected while donor fluorophores are alternatively exited, background fluorescence can be greatly reduced, and thus remarkably higher probe concentrations can be used for single-molecule imaging[18–20]. Compared to the previous techniques, dynamic FRET-FISH has an additional advantage that signals from non-specifically binding DNA probes is inherently eradicated.

To characterize the capabilities of dynamic FRET-FISH, we first used synthetic let-7a as a target. As in Ago-FISH[11], the first step of miRNA detection of dynamic FRET-FISH was the addition of poly (A) tail to the 3′-end of the target miRNA using poly (A) polymerase (Fig. 1b). Biotinylated poly (T) DNA strands were then hybridized to poly (A) RNA tail in a tube. The DNA-RNA hybrid was surface-immobilized on a miRNA detection chamber using biotin-streptavidin interaction.

For single-molecule imaging, we introduced fluorophore-labeled DNA probes into the detection chamber. The Alexa-488-labeled probe (mid probe) and Cy3-labeled probe (tail probe) were complementary to the mid- and tail regions of let-7a miRNA, respectively (Fig. 1c). Their binding sites were designed to be partially overlap each other so that their simultaneous binding did not occur. Cy5-labeled probe (seed probe) was complementary to the seed-region of let-7a miRNA. Binding of all the probes to the target miRNAs was transient because the length of the probes was short in the range of 7 nt and 9 nt. As in SiMREPS and Ago-FISH, the use of short probes was essential to obtain the high specificity of single-nucleotide discrimination. Compared to SiMREPS experiments which were performed at 20 nM probe concentration, and Ago-FISH experiments which were performed at 2 nM probe concentration, however, dynamic FRET-FISH experiments could be performed at higher probe concentrations (donor probe concentration: 1200 nM, acceptor probe concentration: 40 nM) so that clear single-molecule fluorescence time traces of probe binding could be obtained (Fig. 1d and Supplementary Fig. 1). To quantitatively compare the miRNA detection speed of Ago-FISH and FRET-FISH, we determined how long observation time was required to detect half of target molecules (Fig. 1e). Due to the increased probe concentrations, the miRNA detection speed was increased by 60-fold compared to Ago-FISH, and miRNA detection time decreased from 10 min (47.2% relative detection in Ago-FISH) to 10 s (54.0% relative detection in FRET-FISH). The reported miRNA detection speed of SiMREPS at 20 nM probe concentration was similar to that of Ago-FISH at 2 nM probe concentration.

To characterize the target specificity of dynamic FRET-FISH, we prepared three let-7a mutants that had a one-nucleotide mutation in the seed, mid, and tail regions of let-7a, respectively (let-7a_m1, let-7a_m2, and let-7a_m3 in Fig. 2a), and performed dynamic FRET-FISH experiments. Accumulated single-molecule fluorescence images of the wild type and mutant let-7a miRNAs are shown in Fig. 2b: single-molecule spots observed at Alexa-488-excitation only, Cy3 excitation only, and at both excitations are colored in blue, green, and white, respectively. Representative fluorescence time traces used to make the single-molecule images

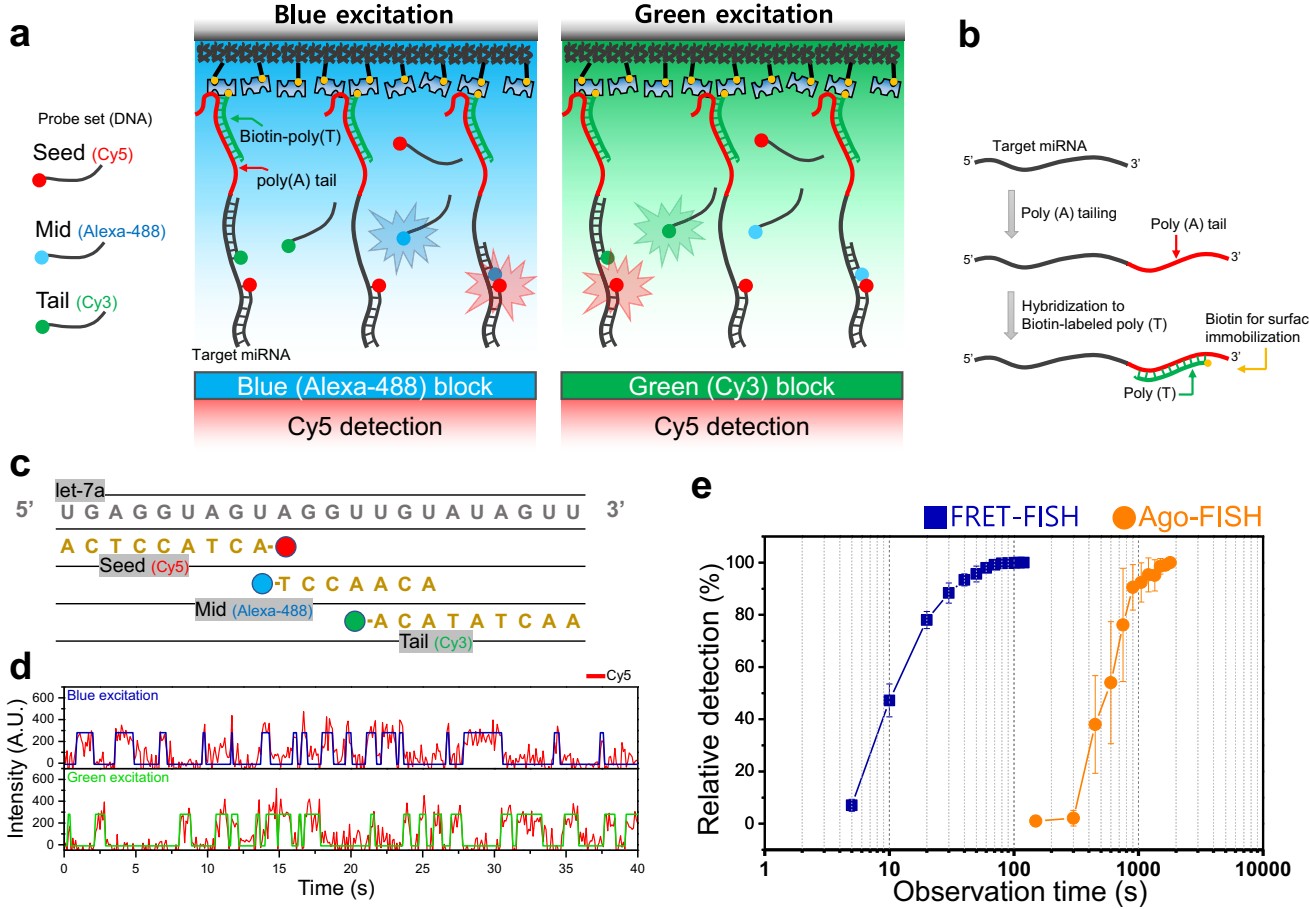

**Fig. 1 miRNA detection using dynamic FRET-FISH. a** Scheme of target miRNA detection. Only the acceptor signal is used for miRNA detection whereas only donor fluorophores (Alexa-488 and Cy3) are excited. In this way, background acceptor fluorescence coming from floating accepter strands is reduced because the acceptor signal is generated only when both donor and acceptor strands simultaneously bind to target miRNAs. **b** miRNA preparation. Purified RNA is hybridized with biotinylated poly (T) after poly (A) tailing. Then RNA is immobilized on a surface via biotin-streptavidin interaction. **c** Sequences of the target miRNA (let-7a), Cy5-labeled seed probe, Alexa-488-labeled mid probe, and Cy3-labeled tail probe. **d** Representative Cy5 fluorescence intensity time traces (red) detected at Alexa-488 excitation (top) and at Cy3 excitation (bottom). The blue and green traces represent the results of hidden Markov modeling of acceptor signals at Alexa-488 and Cy3 excitations, respectively. **e** Observation time dependency of the let-7a number detected using dynamic FRET-FISH (blue) and Ago-FISH (orange). In dynamic FRET-FISH, we identified as a proper target miRNA if a molecule exhibited an acceptor (Cy5) fluorescence signal at both Alexa-488 and Cy3 excitations during the designated observation time. In Ago-FISH, 5% duty cycle of the three DNA probes was used as a threshold for target identification[6]. The duty cycle is defined as the fraction of the total observation time in which a probe was binding to the target miRNA. For dynamic FRET-FISH, 1200 nM donor and 40 nM acceptor probes were used. For Ago-FISH, 2 nM probes were used.

are shown in Fig. 2c. Because the binding of the DNA probes targeting the mutated region was selectively hindered, white spots (molecules that showed the binding of all the three probes) were observed only rarely in all mutant samples, demonstrating that dynamic FRET-FISH can reliably distinguish the wild type let-7a from all its point mutants in the seed, mid, and tail regions.

To benchmark dynamic FRET-FISH against the commercialized PCR techniques, we prepared synthetic let-7 family miRNAs (Fig. 2d) and quantified the true and false positive rates of let-7 family miRNAs (Fig. 2e) using DNA probes optimized for their detection (Supplementary Fig. 2); we identified as a proper target miRNA if a molecule exhibited an acceptor (Cy5) fluorescence signal during the initial 10 s at both Alexa-488 and Cy3 excitations. It is clear that the false positive rates of dynamic FRET-FISH is reduced compared to those of commercial PCR techniques, but similar to those of Ago-FISH[21] (Supplementary Fig. 3).

Finally, we tested dynamic FRET-FISH for the quantification of endogenous miRNAs. Synthetic let-7a was spiked-in total RNA extract of HeLa-S3, and the number of let-7a molecules was counted at varying concentration of the spiked-in let-7a using

dynamic FRET-FISH (black squares, Fig. 3). For comparison, the same experiment was repeated in a buffer with no RNA as a control (red circles, Fig. 3). Although the total RNA sample and control sample were completely different in terms of non-target RNA amounts, the two datasets were well fitted to linear curves with similar slopes, indicating that the same calibration data could be used for quantification of miRNAs in various conditions. Their y-intercepts, however, were different due to the existence of endogenous let-7a in the total RNA sample.

## Discussion

Conventional nucleic acid detection techniques such as microarray, PCR, and NGS do not satisfactorily work for miRNA quantification in terms of specificity and quantification accuracy. A number of amplification-free techniques[13–15] solved the problems of quantification error incurred by polymerase chain reaction (PCR) but still have problems in obtaining high target specificity. SiMREPS and Ago-FISH, recently developed miRNA detection techniques based on single-molecule imaging, solved

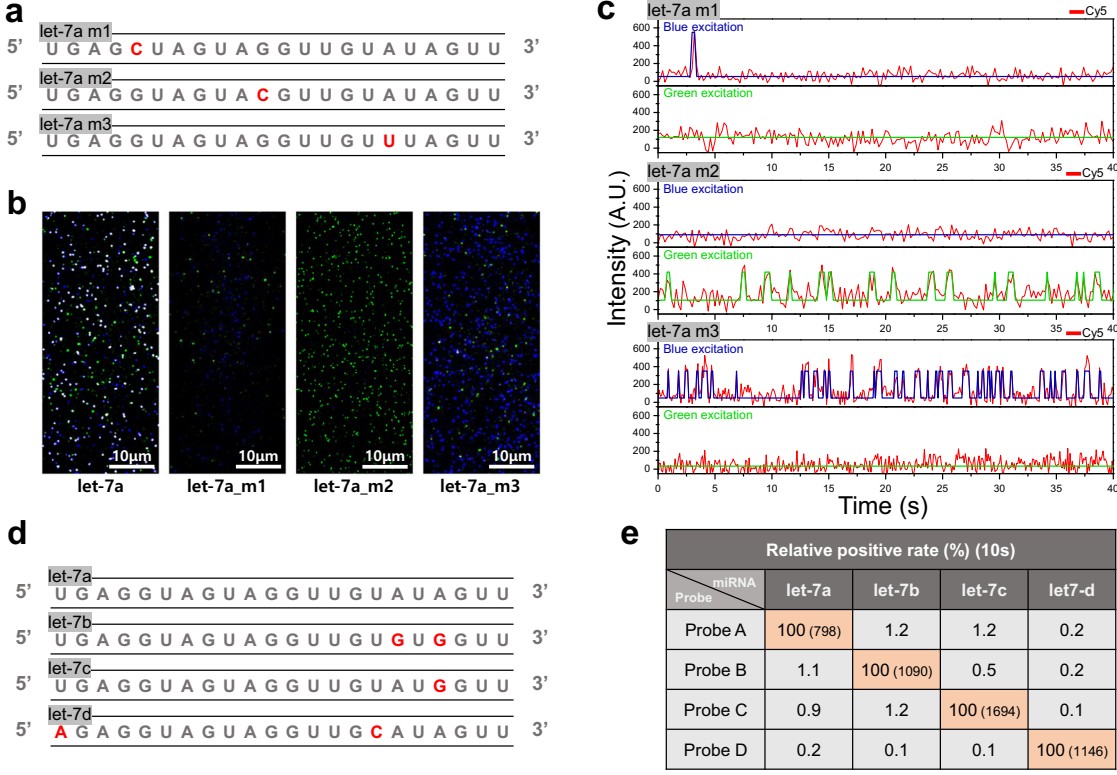

**Fig. 2 Specificity of dynamic FRET-FISH. a** Sequences of let-7a point mutants. Red letters indicate mutation sites. **b** Single-molecule images of wild type let-7a and its point mutants. To generate the images, the acceptor signal was accumulated for 60 s at Alexa-488 excitation (blue) and Cy3 excitation (green). Spots where blue and green colors overlap were colored in white. **c** Representative acceptor fluorescence intensity time traces of let-7a point mutants. **d** Sequences of let-7 family miRNAs. Red letters indicate sequence variations from let-7a. **e** False positive rates of let-7 family miRNA detection. For the experiments, we immobilized 100 pM target miRNAs and injected the designated probe sets. Molecules were identified as a proper target miRNA if an acceptor (Cy5) fluorescence signal occurs during the initial 10 s at both Alexa-488 and Cy3 excitations. The false positive rates were calculated by dividing the off-target number by the on-target number. The Numbers in parentheses are the total molecules that we analysed in the experiments.

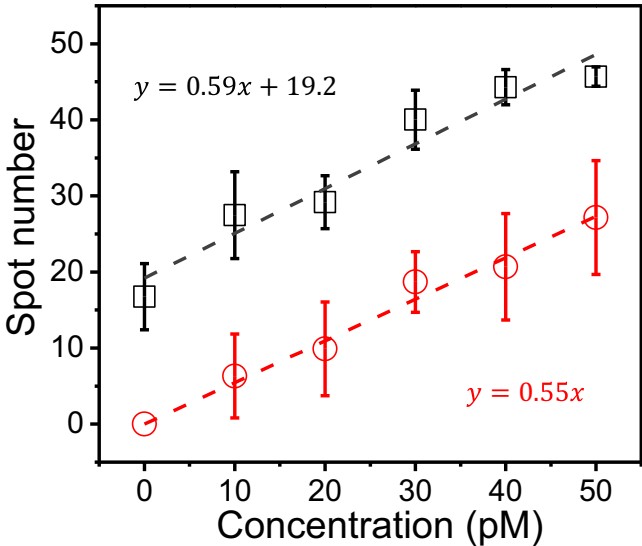

**Fig. 3 Quantification of endogenous miRNAs.** Varying amounts of synthetic let-7a were spiked in the total endogenous RNA of HeLa-S3 cell and the spot number was plotted as a function of spiked let-7a concentration (black squares). The same experiments were performed in a buffer without RNA as a control (red circles).

the problems by providing highly specific and accurate methods of miRNA quantification without target amplification. These techniques, however, are not fast enough; it takes more than 10 min to quantify a single miRNA target[22]. This problem is a serious huddle that should be overcome for rapid profiling of multiple targets. We presented FRET-based miRNA detection technique, dynamic FRET-FISH which allows us to use probe concentration higher than one micromolar. Compared to SiM-REPS and Ago-FISH, its miRNA detection speed is increased by 60-fold while maintaining the advantages of high specificity and accurate quantification of the previous single-molecule miRNA detection techniques. Since it can detect a single miRNA in 10 s, we think that this technique has a potential to be utilized for a rapid diagnosis of early cancer by profiling hundreds of blood circulating miRNAs in an hour. In addition, dynamic FRET-FISH can be used for rapid diagnosis of infectious diseases and imaging because it can be easily converted into a general nucleic acid detection technique[22].

## Methods

**Oligonucleotides and total RNA preparation.** DNA and RNA strands were purchased from Integrated DNA Technology (IDT, Coralville, IA). Amine-modified DNA probes were labeled with Alexa-488, Cy3, or Cy5 mono NHS-esters by incubating 0.05 mM DNA with 3.0 mM fluorophores in a reaction buffer (100 mM $Na_2BO_7$ pH 8.5) for 3 h. The excess dyes were removed by using ethanol

precipitation. Labeled oligonucleotides were stored in distilled water. Total RNA from human HeLa-S3 was purchased from Thermo Fisher Scientific (AM7852).

**Poly (A) tailing of RNA**. Endogenous RNAs (66.7 ng/μl) were incubated for 1 h at 37 °C in a 20 mM Tris-HCl (pH 7.0) buffer with 0.6 mM MnCl$_2$, 20 μM EDTA, 0.2 mM DTT, 100 μg/ml acetylated BSA, 10% glycerol, 3.33 mM rATP, 32 U/μl Yeast Poly (A) Polymerase (Poly (A) Polymerase, Yeast, Thermo Fisher Scientific), and 0.3 U/μl RNase inhibitor (RNasin Plus RNase Inhibitor, Promega). The reaction was terminated by heating the tube for 15 min at 65 °C.

**Single-molecule experiment**. Single-molecule experiments were performed using a total internal reflection fluorescence microscope. A detection chip was made between a quartz slide and a glass coverslip by using double-side sticky tape[23]. To reduce the nonspecific binding of molecules, the surface of the detection chip was coated with a mixture of PEG and biotin-PEG with a 20:1 ratio. 50 ng/μl poly (A)-tailed total endogenous RNA was incubated with 0.45 μM biotinylated poly (T) for 5 min at 55 °C followed by for 3 h at room temperature. After the incubation, 50 μg/μl total RNA was diluted to 16.67 ng/μl using distilled water, and then immobilized on surface by streptavidin–biotin interaction. Then, 1200 nM Alexa-488-labeled mid probe, 1200 nM Cy3-labeled tail probe, and 40 nM Cy5-labeled seed probe in imaging buffer (20 mM Tris-HCl (pH 8.0) with, 400 mM NaCl, 0.5% formamide, 80 mM UREA, and gloxy-mediated oxygen scavenger system) were injected into the detection chamber. Experiments were performed at 30 °C. Alexa-488 and Cy3 were excited by the 473-nm laser and 532-nm laser, respectively. The fluorescence signal was collected through a water-immersion objective (UPlanSApo 60×, Olympus). Alexa-488 and Cy3 induced background noise was eliminated by using a dichroic mirror (635dcxr, Chroma Technology). Finally, Cy5 signal was imaged on an EM-CCD camera (Ixon DV897, Andor). Data were collected by using a home-built program written in LabView (2009, National Instruments). IDL (7.0, ITT), MATLAB (R2013a, The MathWorks), and Origin (8.5, OriginLab) were used for data analysis.

**Statistics and reproducibility**. All data are obtained from more than five experiments and error bars are calculated by standard error of the mean.

**Reporting summary**. Further information on research design is available in the Nature Research Reporting Summary linked to this article.

## Data availability
A reporting summary for this Article is available as a Supplementary Information file. All source data used in graphs in main and supplementary figures are included in Supplementary Data 1. Any other relevant data are available from the corresponding authors upon reasonable request.

## Code availability
The source code used for Hidden Markov Model analysis can be found at https://github.com/ebfret/ebfret-gui [24,25]. Any other custom scripts are available from the corresponding authors upon reasonable request.

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

## Acknowledgements
This work was supported by grants from the National Research Foundation of Korea (2022R1I1A1A01071783 to S.S. and 2022R1A2C3008746 to S.H.).

## Author contributions
S.H. and S.S. conceived the study. S.S., J.K., and C.K. designed and performed the experiments. S.H. and S.S. supervised the study. All authors contributed to writing and revising the paper.

## Competing interests
A US patent application, serial number 10-2019-0037949, was filed by the Seoul National University on technologies described herein.
