## [Peer Review File · Communications Biology]

Reviewers' comments:

Reviewer #1 (Remarks to the Author):

In the present manuscript, Juyoung Kim and co-authors report a method for miRNAs quantification by using dynamic FRET-FISH. After reading the manuscript, I consider that it can be suitable for publication in Communications Biology journal after major revision:

1. In the introduction, the authors could include some examples of "Stable and diagnostically useful miRNAs".
2. The authors should include the meaning of the "SiMREPS and Ago-FISH" abbreviations before their appearance in the text.
3. It could be interesting to include the time of each step of the method (e.g. addition of poly (A) tail to the 3'-end of the target miRNA, ...)
4. The authors could also test the method with lower probes' concentrations (for instance the same used in Ago-FISH), in order to become the method cheaper. Moreover, by decreasing the probes' concentration to those used in Ago-Fish, the method continues to be faster?
5. In the legend of Figure 1 d) and e) the authors should include the probes' concentrations.
6. The authors should include in the supplementary information, the results of Ago-Fish with the mutated sequences and with the let-7 family sequences to demonstrate if their method is better/similar or worst.
7. To prove the biological applicability of the method, the authors should include results with samples of healthy and patients with increased let-7 expression, to show if the method is capable to differentiate samples that are rich in several miRNAs.

Reviewer #2 (Remarks to the Author):

This manuscript "Rapid quantification of miRNAs by using dynamic FRET-FISH" by Kim, J. et al has described a method called FRET-FISH for rapid detection of and quantification of miRNAs. It has been claimed that the detection can be achieved in 10 s at high specificity. While the method has some potential benefit by rapid diagnosis of diseases via miRNA detection, claims regarding speedy detection and specificity have not been fully established (elaborated below in my comments). Mainly, the manuscript does not provide adequate references to give confidence that the claims are genuine. In that respect, I find that this communication has to be significantly revised to support claims before considering for publication.

Major Comments:

-The manuscript narrowly focused on comparing the method presented with SiMREPS and Ago-FISH while several other hybridization-based methods have been published demonstrating amplification-free detection of miRNAs. Therefore, the manuscript in current form did not give me confidence that the claims are fully credible. For example, following two sentences on page 1 and 2: "These techniques, however, are relatively slow: the detection of a single miRNA took more than 10 minutes. Furthermore, the time required for multiple miRNA profiling linearly increases with the number of miRNAs since they should be sequentially quantified."

-Only two example traces are shown as raw data for target detection (Fig. 1d). Several traces should be shown to demonstrate reproducibility of single-molecule behavior. The FRET-traces were fit using the HMM model without providing any kinetic information, which needs to be justified.

-On page 1, it has been claimed that "It is clear that the false positive rates of dynamic FRET-FISH are

significantly reduced compared to those of commercial PCR techniques (10)". Significantly reduced is a relative term. A quantitative estimate is needed.

-On page 3, the author claimed that "Due to the increased probe concentrations, the miRNA detection speed was increased by 60-fold compared to Ago FISH, and miRNA detection time decreased from 10 minutes to 10 s (Figure 1e)". It is not clear where this absolute number "10 minutes" come from. Without full justification, this claim is not acceptable.

-DNA-paint is a similar technique that is well-established for miRNA detection. How the FRET-FISH technique differs from DNA-paint has not been clarified.

-The mutant analysis in fig 2C shows that mutant 2 and mutant 3 are well detected. The authors need to address the following questions regarding this data and related claims: What is the main purpose of this figure if the mutant signals are the same as wild type targets? What happens to the specificity if mutants are detected in a similar manner as the true target?

Other Comments:

-How do the authors know that the biotinylated DNA binds to the end of the Poly(A) tail (as depicted) as the poly(A) tail is longer than the poly(T) strand?

-In Fig. 1 legend, it is not clear what it meant by "5% duty cycle".

-Fig, 2e, what do the numbers in parenthesis in the table represent?

Reviewer #3 (Remarks to the Author):

Hohng and coworkers reported a rapid, highly specific miRNA detection method by decreasing the background noise strategy using FRET. This method has both simpleness and rapidity, showing high potential application.

Our detailed comments are as follows:

1. The background introduction is lack logic, and the authors should give a detailed summary. For example, The introduction about Ago-FISH is not clear and scattered. Authors should give the Ago-FISH information in the front section of the manuscript.

2. Why authors choose Ago-FISH as the comparative method? The detailed compared data with other amplification-free miRNA detection methods should be listed in the manuscript or supporting information.

3. Line 45-48: "This simple scheme, however, could not work because the background noise coming from floating fluorescent probes increased with the probe concentration, overwhelming single-molecule fluorescence signals coming from target-binding probes at probe concentrations higher than tens of nanomolar." Please give references or the detailed analysis.

Response to the Comments of Referee #1:

In the present manuscript, Juyoung Kim and co-authors report a method for miRNAs quantification by using dynamic FRET-FISH. After reading the manuscript, I consider that it can be suitable for publication in Communications Biology journal after major revision.

[Our response] We would like to thank Referee #1 for valuable suggestions to improve the manuscript and tried to address the Reviewer's comments as shown below. We hope the Referee will find the revised manuscript improved enough to be acceptable to *Communications Biology*.

Major Comments:

(1) In the introduction, the authors could include some examples of "Stable and diagnostically useful miRNAs".

[Our response] Following the suggestion, we revised the manuscript as shown below.

"A large number of miRNAs are found to be dysregulated in a broad spectrum of cancers in a disease-specific fashion³. Stable and diagnostically useful miRNAs such as miR-20a^{4, 5} for lung cancer and miR-421^{6, 7} for gastric cancer have been detected in blood, making them ideal biomarkers for early cancer detection^{8, 9}."

(2) The authors should include the meaning of the "SiMREPS and Ago-FISH" abbreviations before their appearance in the text.

[Our response] Following the suggestion, we revised the manuscript as shown below.

"The challenges were recently addressed by amplification-free miRNA detection techniques based on single-molecule fluorescence imaging called SiMREPS (Single Molecule Recognition through Equilibrium Poisson Sampling)¹⁰ and Ago-FISH (Argonaute-based Fluorescence In Situ Hybridization)¹¹, providing reliable ways to accurately profile various miRNAs with highly-specific single-nucleotide discrimination."

(3) It could be interesting to include the time of each step of the method (e.g. addition of poly (A) tail to the 3'-end of the target miRNA, ...).

[Our response] The information was included in the Methods section.

(4) The authors could also test the method with lower probes' concentrations (for instance the same used in Ago-FISH), in order to become the method cheaper. Moreover, by decreasing the probes' concentration to those used in Ago-Fish, the method continues to be faster?

[Our response] As explained in Introduction, the target detection speed of FRET-FISH decreases with probe concentrations. For example, at 2 nM probe concentration, which was used for Ago-FISH, FRET signal is too rare to be practically used for miRNA detection since the binding rate of FRET probes decreases by 500-fold.

(5) In the legend of Figure 1 d) and e) the authors should include the probes' concentrations.

[Our response] The information is included in the Figure legend now.

(6) The authors should include in the supplementary information, the results of Ago-Fish with the mutated

sequences and with the let-7 family sequences to demonstrate if their method is better/similar or worst.

[Our response] Following the suggestion, we included the information in the Supplementary Data (Supplementary Fig. 3).

(7) To prove the biological applicability of the method, the authors should include results with samples of healthy and patients with increased let-7 expression, to show if the method is capable to differentiate samples that are rich in several miRNAs.

[Our response] At current stage, we are afraid that the suggested experiments cannot be performed; acquiring human plasma samples requires lots of documental works and time. We will address the issue in the following up-up studies.

Response to the Comments of Referee #2:

This manuscript "Rapid quantification of miRNAs by using dynamic FRET-FISH" by Kim, J. et al has described a method called FRET-FISH for rapid detection of and quantification of miRNAs. It has been claimed that the detection can be achieved in 10 s at high specificity. While the method has some potential benefit by rapid diagnosis of diseases via miRNA detection, claims regarding speedy detection and specificity have not been fully established (elaborated below in my comments). Mainly, the manuscript does not provide adequate references to give confidence that the claims are genuine. In that respect, I find that this communication has to be significantly revised to support claims before considering for publication.

[Our response] We would like to thank Referee #2 for helpful suggestions to improve the manuscript. We hope the Referee #2 will find the revised manuscript improved enough to be acceptable to *Communications Biology*.

Major Comments:

(1) The manuscript narrowly focused on comparing the method presented with SiMREPS and Ago-FISH while several other hybridization-based methods have been published demonstrating amplification-free detection of miRNAs. Therefore, the manuscript in current form did not give me confidence that the claims are fully credible. For example, following two sentences on page 1 and 2: "These techniques, however, are relatively slow: the detection of a single miRNA took more than 10 minutes. Furthermore, the time required for multiple miRNA profiling linearly increases with the number of miRNAs since they should be sequentially quantified."

[Our response] The SiMREPS and Ago-FISH are distinguished from other hybrid-based techniques for miRNA detection in that they detect transient interactions between target and probes, and thus superior to other techniques in terms of target specificity. To clarify the point, we extended the Introduction section as follows.

"The challenges were recently addressed by amplification-free miRNA detection techniques based on single-molecule fluorescence imaging called SiMREPS (Single Molecule Recognition through Equilibrium Poisson Sampling)¹⁰ and Ago-FISH (Argonaute-based Fluorescence In Situ Hybridization)¹¹, providing reliable ways to accurately profile various miRNAs with highly-specific single-nucleotide discrimination. SiMREPS, a miRNA detection method inspired by the super-resolution imaging technique, DNA-PAINT (DNA-based point accumulation for imaging in nanoscale topography)¹², used the transient binding of fluorescently-labeled short DNA probes to surface-immobilized target miRNAs for miRNA detection, and demonstrated a highly-confident miRNA detection based on binding kinetics analysis with high specificity of 500-fold discrimination between single nucleotide polymorphisms. Ago-FISH, another miRNA detection technique inspired by DNA-PAINT, uses the transient binding of Argonaute-loaded DNA probes instead of bare DNA probes to obtain 20-fold increased probe binding rate compared to SiMREPS at the same probe concentrations while maintaining the similar high target specificity of SiMREPS. The quantification accuracies of SiMREPS and Ago-FISH were also superior to the conventional techniques such as NGS and PCR because no nonlinear amplification step was required for miRNA quantification. Recently other amplification-free miRNA detection techniques based on target-probe hybridization have been developed¹³⁻¹⁵, but they could not demonstrate such a high target specificity of SiMREPS and Ago-FISH because the techniques required a stable binding of probes to target miRNAs like the

conventional microarray technique. A potential disadvantage of SiMREPS and Ago-FISH, however, is that they should detect different miRNAs sequentially, making the time required for multiple miRNA profiling linearly increase with the number of profiled miRNAs. Although SiMREPS and Ago-FISH can detect a single miRNA much faster (10 minutes) than PCR (several hours), this advantage disappears when hundreds of miRNAs should be profiled at once.”

(2) Only two example traces are shown as raw data for target detection (Fig. 1d). Several traces should be shown to demonstrate reproducibility of single-molecule behavior. The FRET-traces were fit using the HMM model without providing any kinetic information, which needs to be justified.

[Our response] Following the suggestion, more traces and the information of binding kinetics obtained from **more than 3,000 molecules** is included in Supplementary Fig. 2.

(3) On page 1, it has been claimed that “It is clear that the false positive rates of dynamic FRET-FISH are significantly reduced compared to those of commercial PCR techniques (18)”. Significantly reduced is a relative term. A quantitative estimate is needed.

[Our response] The information can be found in the reference **#18**. For the convenience of the readers, however, the information is included in Supplementary Fig. 3 now.

(4) On page 3, the author claimed that “Due to the increased probe concentrations, the miRNA detection speed was increased by 60-fold compared to Ago FISH, and miRNA detection time decreased from 10 minutes to 10 s (Figure 1e)”. It is not clear where this absolute number “10 minutes” come from. Without full justification, this claim is not acceptable.

[Our response] To clarify the criteria that we used to determine the miRNA detection speeds of Ago-FISH and FRET-FISH, we revised the manuscript as follows.

“To quantitatively compare the miRNA detection speed of Ago-FISH and FRET-FISH, we determined how long observation time was required to detect half of target molecules (Fig. 1e). Due to the increased probe concentrations, the miRNA detection speed was increased by 60-fold compared to Ago-FISH, and miRNA detection time decreased from 10 minutes (47.2 % relative detection in Ago-FISH) to 10 s (54.0 % relative detection in FRET-FISH). The reported miRNA detection speed of SiMREPS at 20 nM probe concentration was similar to that of Ago-FISH at 2nM probe concentration.”

(5) DNA-paint is a similar technique that is well-established for miRNA detection. How the FRET-FISH technique differs from DNA-paint has not been clarified.

[Our response] DNA-PAINT is a superresolution imaging technique, not a miRNA detection technique. SiMREPS and Ago-FISH were inspired by DNA-PAINT, and developed to specifically detect miRNAs. To clarify the point, we added the following sentences to the Introduction section.

“SiMREPS, a miRNA detection method inspired by the super-resolution imaging technique, DNA-PAINT (DNA-based point accumulation for imaging in nanoscale topography)¹², used the transient binding of fluorescently-labeled short DNA probes to surface-immobilized target miRNAs for miRNA detection, and demonstrated a highly-confident miRNA detection based on binding kinetics analysis with high specificity of 500-fold discrimination between single nucleotide polymorphisms. Ago-FISH, another miRNA detection technique inspired by DNA-PAINT, uses the transient binding of Argonaute-

loaded DNA probes instead of bare DNA probes to obtain 20-fold increased probe binding rate compared to SiMREPS at the same probe concentrations while maintaining the similar high target specificity of SiMREPS.”

(6) The mutant analysis in fig 2C shows that mutant 2 and mutant 3 are well detected. The authors need to address the following questions regarding this data and related claims: What is the main purpose of this figure if the mutant signals are the same as wild type targets? What happens to the specificity if mutants are detected in a similar manner as the true target?

[Our response] We think the Reviewer misunderstood this part. We identified a molecule as a true target when the acceptor signal was detected at both blue and green excitations as shown in Fig. 1d. This point was already explained in the manuscript as shown below.

“Representative fluorescence time traces used to make the single-molecule images are shown in Figure 2c. Because the binding of the DNA probes targeting the mutated region was significantly and selectively hindered, white spots (molecules that showed the binding of all the three probes) were observed only rarely in all mutant samples, demonstrating that dynamic FRET-FISH can reliably distinguish the wild type let-7a from all its point mutants in the seed, mid, and tail regions.”

Other Comments:

(1) How do the authors know that the biotinylated DNA binds to the end of the Poly(A) tail (as depicted) as the poly(A) tail is longer than the poly(T) strand?

[Our response] As the reviewer pointed out, the binding position is random. To clarify the point, we revised Figure 1.

(2) In Fig. 1 legend, it is not clear what it meant by “5% duty cycle”.

[Our response] To clarify the meaning of the duty cycle, we added the following sentence to the Figure caption.

“The duty cycle is defined as the fraction of the total observation time in which a probe was binding to the target miRNA.”

(3) Fig, 2e, what do the numbers in parenthesis in the table represent?

[Our response] The Numbers in parentheses are the total molecules we analyzed in the experiments. To clarify the point, we revised the Figure legend.

Response to the Comments of Referee #3:

Hohng and coworkers reported a rapid, highly specific miRNA detection method by decreasing the background noise strategy using FRET. This method has both simpleness and rapidity, showing high potential application. Our detailed comments are as follows:

[Our response] We would like to thank Referee #3 for several valuable suggestions to improve the manuscript. We hope the Referee #3 will find the revised manuscript improved enough to be acceptable to *Communications Biology*.

Major Comments:

(1) The background introduction is lack logic, and the authors should give a detailed summary. For example, the introduction about Ago-FISH is not clear and scattered. Authors should give the Ago-FISH information in the front section of the manuscript.

[Our response] Following the suggestion, we revised the manuscript as follows.

“**The challenges were recently addressed by** amplification-free miRNA detection techniques based on single-molecule fluorescence imaging called **SiMREPS (Single Molecule Recognition through Equilibrium Poisson Sampling)¹⁰** and **Ago-FISH (Argonaute-based Fluorescence In Situ Hybridization)¹¹**, providing reliable ways to accurately profile various miRNAs with **highly-specific single-nucleotide discrimination**. **SiMREPS, a miRNA detection method inspired by the super-resolution imaging technique, DNA-PAINT (DNA-based point accumulation for imaging in nanoscale topography)¹²**, used the transient binding of fluorescently-labeled short DNA probes to surface-immobilized target miRNAs for miRNA detection, and demonstrated a highly-confident miRNA detection based on binding kinetics analysis with high specificity of 500-fold discrimination between single nucleotide polymorphisms. Ago-FISH, another miRNA detection technique inspired by DNA-PAINT, uses the transient binding of Argonaute-loaded DNA probes instead of bare DNA probes to obtain 20-fold increased probe binding rate compared to SiMREPS at the same probe concentrations while maintaining the similar high target specificity of SiMREPS. The quantification accuracies of SiMREPS and Ago-FISH were also superior to the conventional techniques such as NGS and PCR because no nonlinear amplification step was required for miRNA quantification. Recently other amplification-free miRNA detection techniques based on target-probe hybridization have been developed¹³⁻¹⁵, but they could not demonstrate such a high target specificity of SiMREPS and Ago-FISH because the techniques required a stable binding of probes to target miRNAs like the conventional microarray technique. A potential disadvantage of SiMREPS and Ago-FISH, however, is that they should detect different miRNAs sequentially, making the time required for multiple miRNA profiling linearly increase with the number of profiled miRNAs. Although SiMREPS and Ago-FISH can detect a single miRNA much faster (10 minutes) than PCR (several hours), this advantage disappears when hundreds of miRNAs should be profiled at once.”

(2) Why authors choose Ago-FISH as the comparative method? The detailed compared data with other amplification-free miRNA detection methods should be listed in the manuscript or supporting information.

[Our response] Please refer to our answer to the previous comment.

(3) Line 45-48: "This simple scheme, however, could not work because the background noise coming from floating fluorescent probes increased with the probe concentration, overwhelming single-molecule fluorescence signals coming from target-binding probes at probe concentrations higher than tens of nanomolar." Please give references or the detailed analysis.

[Our response] Following the suggestion, we added references to the revised manuscript.

REVIEWERS' COMMENTS:

Reviewer #1 (Remarks to the Author):

The authors have addressed some of my concerns. I ask the authors only to comment in the main manuscript the result of Fig 3 of SI.

Thank you!

Reviewer #2 (Remarks to the Author):

The authors have addressed my concerns in the revised manuscript. After revision, the quality of the manuscript has been much improved.

Response to the Comments of Referee #1:

The authors have addressed some of my concerns. I ask the authors only to comment in the main manuscript the result of Fig 3 of SI. Thank you!

[Our response] As suggested, Fig. S3 is referred in the main manuscript as shown below.

“It is clear that the false positive rates of dynamic FRET-FISH is reduced compared to those of commercial PCR techniques, but similar to those of Ago-FISH¹ (Supplementary Fig. 3).”

Response to the Comments of Referee #2:

The authors have addressed my concerns in the revised manuscript. After revision, the quality of the manuscript has been much improved.

[Our response] We would like to thank Referee #2 for having provided valuable suggestions to improve our manuscript.